# Retrieving Ground-Level PM$_{2.5}$ Concentrations in China (2013-2021) with a Numerical Model-Informed Testbed to Mitigate Sample Imbalance-Induced Biases

Siwei Li [1,3,4], Yu Ding[1], Jia Xing[2], Joshua S. Fu[2]

[1]Hubei Key Laboratory of Quantitative Remote Sensing of Land and Atmosphere, School of Remote Sensing and Information Engineering, Wuhan University, Hubei, 430000, China
[2]Department of Civil and Environmental Engineering, the University of Tennessee, Knoxville, TN 37996, USA
[3]State Key Laboratory of Information Engineering in Surveying, Mapping and Remote Sensing, Wuhan University, Wuhan 430079, China
[4] Hubei Luojia Laboratory, Wuhan University, Wuhan 430079, China

*Correspondence to*: Siwei Li (siwei.li@whu.edu.cn); Jia Xing (jxing3@utk.edu)

**Abstract.** Ground-level PM$_{2.5}$ data derived from satellites with machine learning are crucial for health and climate
assessments, however, uncertainties persist due to the absence of spatially covered observations. To address this, we propose a novel testbed using untraditional numerical simulations to evaluate PM$_{2.5}$ estimation across the entire spatial domain. The testbed emulates the general machine-learning approach, by training the model with grids corresponding to ground monitor sites and subsequently testing its predictive accuracy for other locations. Our approach enables comprehensive evaluation of various machine-learning methods' performance in estimating PM$_{2.5}$ across the spatial domain for the first time. Unexpected
results are shown in the application in China, with larger absolute PM$_{2.5}$ biases found in densely populated regions with abundant ground observations across all benchmark models due to the higher baseline concentration, though relative errors (approximately 20%) is smaller compared to rural areas (over 50%). The imbalance in training samples, mostly from urban areas with high emissions, is the main reason, leading to significant overestimation due to the lack of monitors in downwind areas where PM$_{2.5}$ is transported from urban areas with varying vertical profiles. Our proposed testbed also provides an
efficient strategy for optimizing model structure or training samples to enhance satellite-retrieval model performance. Integration of spatiotemporal features, especially with CNN-based deep-learning approaches like the ResNet model, successfully mitigates PM$_{2.5}$ overestimation (by 5-30 µg m$^{-3}$) and corresponding exposure (by 3 million people • µg m$^{-3}$) in the downwind area over the past nine years (2013-2021) compared to the traditional approach. Furthermore, the incorporation of 600 strategically positioned ground-measurement sites identified through the testbed is essential to achieve
a more balanced distribution of training samples, thereby ensuring precise PM$_{2.5}$ estimation and facilitating the assessment of associated impacts in China. In addition to presenting the retrieved surface PM$_{2.5}$ concentrations in China from 2013 to 2021, this study provides a testbed dataset derived from physical modeling simulations which can serve to evaluate the performance of data-driven methodologies, such as machine learning, in estimating spatial PM$_{2.5}$ concentrations for the community.

## 1 Introduction


Accurate knowledge of PM$_{2.5}$ pollution is vital for understanding its impact on human health (Lelieveld et al., 2015; Geng et al., 2021) and the climate (Mitchell et al., 1995; Bellouin et al., 2005). Satellite products provide direct measurements of aerosol loading on broad spatial and temporal scales. While, the Aerosol Optical Depth (AOD) measured by satellites reflects the total column of particular matters, challenged by the complex relationship between AOD with ground PM$_{2.5}$

influenced by various factors (Hoff et al., 2009), including aerosol chemical composition and vertical profiles. Compared to traditional statistics, machine learning excels in addressing non-linearities. Therefore, numerous recent studies leverage machine learning, such as Random Forest (RF) (Hu et al., 2017), XGBoost (Xiao et al., 2018), lightGBM (Zhong et al., 2021), and deep learning models (Li et al., 2020; Yan et al., 2020; Wang et al., 2022a; Wang et al., 2022b; Wei et al., 2023) to establish correlations between AOD and PM$_{2.5}$, treating AOD and related factors, including meteorological variables, as

features to predict surface PM$_{2.5}$ based on ground measurements (Ma et al., 2022). However, a limitation arises as most ground measurements are concentrated in urban and polluted areas. Their main purpose is to monitor the high pollution level to protect human health, leading to an uneven spatial distribution. It is expected that training models predominantly on urban sites introduce an imbalance in ground-based measurements, resulting in significant uncertainties in spatially allocating surface PM$_{2.5}$ based on satellite AOD (Shin et al., 2020). This deficiency might be particularly notable in suburban areas

experiencing downwind transport of PM$_{2.5}$ from urban areas (Bai et al., 2022). The discrepancy between urban and downwind sites largely lies in their vertical profiles of aerosol across the entire vertical layers. Urban sites, which have abundant emission sources such as residential areas, transportation, construction, and industries, exhibit a higher share of ground-level aerosol relative to the total AOD compared to downwind and rural areas, where pollution tends to be lifted to upper layers through atmospheric dynamics. Accurately representing the varying aerosol vertical profiles in source/urban

and downwind/rural areas is crucial for retrieving ground-level PM$_{2.5}$ from the AOD. However, imbalanced training samples make the machine-learning model unable to adequately capture such variations. The traditional cross-validation methods either based on samples or sites (Dong et al., 2020), which still rely mostly on samples available in urban sites, fails to provide a comprehensive assessment of model performance across the entire prediction space. Consequently, uncertainties in PM$_{2.5}$ estimation for these areas remain unexplored, and solutions to reduce such uncertainties are yet to be developed.

To overcome these limitations, we introduce a novel testbed utilizing a numerical model, specifically a chemical transport model (CTM) such as the Community Multiscale Air Quality Modeling System developed by the U.S. EPA and its community (Appel et al., 2013), to establish ground-truth data beyond monitor points. This allows for the evaluation of interpolation performance using various machine-learning models and provides solutions to mitigate the uncertainties stemming from the sample imbalance problem. More specifically, we emulate traditional machine-learning methods but by

using CTM-simulated PM$_{2.5}$ concentrations in grid cells corresponding to ground monitor sites as labels for training machine-learning models. Subsequently, we validate the trained model's performance in predicting PM$_{2.5}$ concentrations in other grid cells. In addition to providing a "ground-truth" for assessing performance across the entire space, the CTM

simulated data acts as a testbed for efficiently seeking solutions to enhance satellite-retrieval model performance. This involves optimizing features, model structure, and training samples, as depicted in Figure 1.

## 2 Methods

The proposed testbed is implemented in a China domain, utilizing one whole year simulation of 2017 with a 27km by 27km resolution. To ensure internal consistency during training, all the feature and label data are derived from the input and output of a commonly used CMAQ. The meteorological variables include U-wind, V-wind, humidity, 2-meter temperature, convective velocity scale, short-wave radiation, 10-meter wind speed, PBL height, leaf-area-index (LAI), cloud fraction, and precipitation, simulated by the Weather Research & Forecasting Model (WRF) (Skamarock et al., 2008). The simulated Aerosol Optical Depth (AOD) is calculated based on simulated $PM_{2.5}$ chemical compositions and corresponding meteorological variables across all vertical layers (Liu et al., 2010). We also include the $NO_2$ column density as an important feature as it is highly correlated with emission sources and can be directly observed from satellites to better represent the emission information (Martin et al., 2003). Similarly, the simulated $NO_2$ column density is calculated based on simulated $NO_2$ concentrations and corresponding meteorological variables across all vertical layers.

In addition, we conduct model training using nine-year observational data from 2013 to 2021 to evaluate potential biases under real-world conditions. This is quantified by measuring the difference in retrieved $PM_{2.5}$ between the traditional model and the improved retrieval method optimized with the proposed testbed. The dataset for this evaluation comprises the Moderate Resolution Imaging Spectroradiometer (MODIS) (Remer et al., 2008) satellite observations for AOD, Ozone Monitoring Instrument (OMI) (Celarier et al., 2008) satellite data for $NO_2$ column density, and ground monitor observations of $PM_{2.5}$ from the China National Environmental Monitoring Center (CNEMC, coving more than 600 grid cells of 27km by 27km in total) (Kong et al., 2021). Following the same data filling method (He et al., 2020), we conduct data filling for the satellite measurement of $NO_2$ column density and AOD when applying our approach to real data. The generation of testbed data and the machine learning methods are detailed as follows.

### 2.1 Numerical model WRF/CMAQ

In this study, we utilized version 5.2 of the Community Multiscale Air Quality (CMAQ) model (Appel et al., 2018), incorporating the Carbon Bond 6 (Yarwood et al., 2010) gas-phase chemistry mechanism and the AERO6 particulate matter chemistry mechanism. CMAQ, a widely recognized CTM, is renowned for its accurate simulation of air pollutant concentrations, including $PM_{2.5}$, attributed to its comprehensive representation of particulate matter formations. Meteorological data were generated using the Weather Research and Forecasting (WRF) model, version 3.8, configured in the same manner as our previous studies (Ding et al., 2019ab). Emission data were obtained from the high-resolution emission inventory developed by Tsinghua University (ABaCAS-EI) (Zheng et al., 2019), characterized by a spatial resolution as 27km-by-27km and a temporal resolution of 1 hour to match with the CMAQ model. Biogenic emissions were

derived from the estimation of the Model for Emissions of Gases and Aerosols from Nature (MEGAN) (Guenther et al., 2012). We conducted a thorough assessment of the performance of WRF and CMAQ in simulating meteorological variables and air pollutant concentrations, employing extensive comparisons with observational data in our previous studies (Ding et al., 2019ab).

The simulation domain spans a significant portion of East Asia and is depicted by a grid consisting of 182 rows and 232 columns, featuring a horizontal resolution of 27 km by 27 km. The entirety of the troposphere (from ground level to 100mb) is represented using 14 layers with sigma values, namely 1.00, 0.995, 0.99, 0.98, 0.96, 0.94, 0.91, 0.86, 0.8, 0.74, 0.65, 0.55, 0.4, 0.2, and 0.00. These sigma values correspond to altitudes of 19, 57, 114, 230, 386, 584, 910, 1375, 1908, 2618, 3598, 5061, 7620, and 11944 meters above the ground level, both at the domain and on an annual averaged basis.

We align the simulated $PM_{2.5}$ concentrations from CMAQ with the CNEMC based on their respective locations, treating them as the "label" for training the machine-learning model. Though previous studies provide some validation schemes to evaluate the model's extrapolation capacity (Dong et al., 2020), for the remaining grid cells that encompass surrounding $PM_{2.5}$ areas where observations are not available, the predicted concentrations with machine learning method cannot be directly compared and examined. This paper focuses on assessing the model's performance in predicting these points, accounting for over 90% of the total grid cells. The simulation data serves as the "ground truth" for the evaluation the output of the machine-learning model.

The WRF/CMAQ simulations were evaluated in our previous studies (Ding et al., 2019ab), demonstrating acceptable agreement with CNEMC observations, albeit with limitations in areas where observations are available. In rural areas where no observations are available, direct comparison of CMAQ predictions with actual observations is not possible. However, the CMAQ data used in this study primarily serves to establish a testbed representing scenarios based on physical laws such as emission, diffusion, advection, and deposition. This approach contrasts with reanalysis or data fusion methods, which may deviate from these physical functions, even though they might exhibit better agreement with observations when available.

## 2.2 Decision tree-based machine learning method

This study employed three decision-tree-based machine learning algorithms, namely Random Forest (Belgiu and Drăguţ, 2016), XGBoost (Chen and Guestrin, 2016), and LightGBM (Ke et al., 2017), to serve as benchmark cases, given their widespread use in previous studies. Additionally, Deep Forest (noted as DeepRF in this study) (Zhou et al., 2019), known for its superior performance (Wei et al., 2023), was included as an additional method to be evaluated in this study.

We incorporated similar features used in the machine-learning model, including observed meteorological variables (WRF output) and land use information. The reason is we deliberately avoided using CTM simulation results for two key reasons, while some previous studies included CTM modeling results as additional features in training machine learning models. First, the CTM will be applied to the testbed to evaluate the model's performance, and introducing CTM results could leak information as these results are utilized as labels and therefore cannot be used as input thereafter. Second, we aimed to

propose a comprehensive CTM-free method that relies exclusively on satellite products and meteorological variations obtained from observations. This choice is motivated by the low efficiency of conducting CTM and the uncertainties it introduces. Furthermore, the only additional information provided by the CTM is related to emissions, which still suffers

from uncertainties. Therefore, instead of relying on CTM or prior emission data, we introduce the $NO_2$ column density. This variable is highly correlated with emission sources and can be directly observed from satellites, offering a more accurate representation of emission information.

Given our objective to assess grid cells outside the designated "label", there is no overlap between the training and test datasets. To evaluate the model's performance on the labels, we employ temporal validation. Specifically, the model is

trained using data from only the first 25 days of each month, and the remaining days are reserved for testing. This approach helps gauge the model's effectiveness in handling temporal variations and provides a robust assessment of its performance on the specified labels. We fixed the days for training rather than selecting them randomly to ensure that all methods use exactly the same data for training and testing, enabling a fair comparison. Random selection would still require fixing the randomly selected days for all methods, similar to fixing all days at the outset. Moreover, the purpose of this study is to

investigate prediction errors for grid cells not included in the training dataset. Even when using the first 25 days of training dataset, we consistently observe similar prediction errors in other sites (similar to out-of-sample validation), regardless of which days are selected for training or testing.

## 2.3 Residual neural network method (ResNet)

The incorporation of spatiotemporal-neighborhood features is crucial for enhancing the model's capability to discern the

evolution of vertical profiles in both urban and downwind areas (Chen et al., 2023). Beyond simply including corresponding features from the surrounding neighborhood grid cells as additional predictors for predicting $PM_{2.5}$ concentration at the target grid cells in decision tree-based methods, we also employ a deep-learning method, which is ResNet (He et al., 2016). This choice is motivated by its demonstrated advantage in handling the nonlinearity inherent in atmospheric processes, as suggested in our previous study (Xing et al., 2020).

The ResNet consists of an initial layer with 128 channels and incorporates 8 residual blocks. The feature maps, encompassing meteorological variables, land use information, and AOD, are fed into the Conventional Neural Network (CNN)-based structure with a 3 by 3 kernel size, as illustrated in Figure S1. Additionally, we also incorporate the feature in the previous and next day as additional features to help capture the transport flow of pollutants in the model. The training loss will concentrate solely on points corresponding to the monitor sites, generating predictions exclusively for these specific

locations, given the scattered nature of the labels. As a result, predictions for other points will be entirely out-of-sample, relying on data from the same locations as the monitor sites.

One thing should be noted that all machine learning methods use the same input features to ensure a fair comparison. Only difference is that the features for the new proposed methods (e.g., ResNet) include data from the neighborhood (nearby grid cells and previous/next time-steps) in addition to the local grid and time data.

Throughout the training phase, we employed the Mean Squared Error (MSE) loss function, conducting a total of 1000 epochs, which demonstrated sufficient effectiveness in achieving robust performance during both training and testing. The learning rate started at 0.0001 and underwent linear decay, reaching zero by the conclusion of the training process. Additionally, we utilized the Adam optimizer (Kingma and Ba, 2014) to enhance the convergence of the model.

## 3 Results

### 3.1 Imbalance in site distribution leads traditional methods to overestimate downwind PM$_{2.5}$


To explore uncertainties in traditional machine-learning methods, we initially adhere to their typical design, relying exclusively on local features within each grid cell. This approach involves utilizing only the feature data from the same location as the target grid cell. The trained model with the Random Forest method (RF) successfully captures the spatial distribution of PM$_{2.5}$, showing elevated levels in east China and lower levels in the west (Figure 2a). It exhibits acceptable

performance for the "label" grid cells during validation (Figure 2b, R$^2$=0.98 and RMSE=5.28 µg m$^{-3}$ in the training dataset, and R$^2$=0.81 and RMSE=16.1 µg m$^{-3}$ in the test dataset).

However, considerable errors are observed across space, particularly in polluted regions with high baseline PM$_{2.5}$ concentrations (Figure 2c). Positive biases (i.e., predictions greater than CMAQ) increase with the distance from monitor sites, even as PM$_{2.5}$ concentrations decrease. This suggests an overestimation in predictions for downwind areas away from

the monitor sites (Figure 2d). This is mainly attributed to the traditional model's difficulty in discerning variations in vertical profiles between urban and suburban areas. Training is primarily focused on urban areas, where pollution is concentrated near the surface due to ground-level emission sources. In contrast, pollution in downwind areas is transported aloft. Therefore, the model, trained on urban sites where ground-level pollution from AOD is more prominent, failed to accurately capture the diverse aerosol vertical profiles in source/urban and downwind/rural areas. This discrepancy resulted in

overestimations in downwind areas. (as illustrated in Figure 2e).

Contrary to traditional expectations, significant absolute errors occur mostly in East China rather than in the West due to the large baseline concentration, even though the relative error is smaller (about 20%) (Figure S2). While the east has more densely located monitor sites, they are primarily situated in urban centers. This imbalance in site distribution, combined with much higher concentrations, results in substantial biases in east China.

Similar phenomena are observed in three other benchmark models that have been applied in previous studies, specifically XgBoost, LightGBM, and the deep-RF. All of these models demonstrate robust performance in both training and testing at the monitor sites (R$^2$>0.8 and RMSE<16.2 µg m$^{-3}$ in the test cases, as depicted in Figure S3). However, they display similar uncertainties in downwind PM$_{2.5}$, with significant errors occurring in the surrounding grid cells of the monitor sites rather than in remote sites where concentrations are relatively low. Clearly, we can conclude that the uneven distribution of sites

introduces considerable biases in PM$_{2.5}$ estimation within traditional methods that rely on local features.

**3.2 Inclusion of spatiotemporal-neighbourhood features improves surrounding PM₂.₅ prediction other than traditional approaches**

As previously discussed, the ineffectiveness of a machine-learning model trained on imbalanced samples can be attributed primarily to insufficient information regarding the spatial variation of vertical profiles from the source to the downwind area. To enhance the integration of crucial information regarding vertical profiles, we introduce spatiotemporal-neighbourhood features into the model. This addition aims to empower the model with the capability to distinguish between urban and downwind areas. Leveraging the convolutional neural network (CNN) -based structure, known for its effectiveness in exploring non-linear relationships among neighbouring grid cells, we opt for the widely used deep-learning model ResNet. This choice facilitates the establishment of non-linear relationships between predicted PM₂.₅ concentrations and multiple spatiotemporal-neighbourhood features. In contrast to traditional methods that solely focus on single-time features, our approach incorporates both preceding and succeeding time features to enhance the model's capacity to discern differences among urban and downwind grid cells. This inclusion is motivated by the fact that plume transport is predominantly influenced by flow dynamics, represented by the variation in the temporal neighbourhood (before and after) features. Our previous studies have also demonstrated the effectiveness of linking grid cells with time-series information in PM₂.₅ estimation, underscoring the rationale for this inclusive approach (Teng et al., 2023; Ding et al., 2024). Additionally, considering that AOD measured by satellites, such as MODIS, captures only a single time step while predictions are made for daily averages, the inclusion of extra time-step information proves beneficial in capturing a broader temporal context compared to a single time snapshot (the model noted as ResNet-time).

The results indicate that while the spatial pattern of PM₂.₅ predicted with ResNet closely resembles that of other models (Figure 3a), it significantly enhances model performance in predicting PM₂.₅ for both the training dataset (reducing RMSE from 4 to 2 µg m⁻³ and increasing R² slightly) and the test dataset (reducing RMSE from 14 to 8 µg m⁻³ and increasing R² from 0.8 to 0.9). This improvement can be attributed to the incorporation of both spatial and temporal features. The performance of the traditional RF model is enhanced by replacing it with the ResNet model, and this improvement is further amplified by including temporal features (previous and next-time steps) in the ResNet-time model (see Figure 3b). The incorporation of surrounding features in the ResNet-time model significantly mitigates both absolute and relative errors in East China across the spatial domain (see Figure 3c; Figure S2). However, some deterioration is observed in the West, primarily attributable to limited samples. The model, becoming more complex, lacks sufficient training samples in the West, leading to overfitting in that region.

The inclusion of spatiotemporal-neighborhood features also significantly improves the performance of traditional benchmark models, by incorporating corresponding features of the surrounding eight neighborhood grid cells and the temporal neighbourhood (before and after) information as additional predictors for predicting PM₂.₅ concentration at the target grid cells. Improvements are observed in both training and test datasets across all four benchmark models, as depicted in Figure S4. Notably, all models demonstrate a reduction in RMSE after integrating spatiotemporal-neighborhood features, especially for the downwind area (within the distance of 1-3 grid cells) (see Figure 3d). However, performance is barely improved or

even worsens in faraway sites (distance >4 grid cells) due to the limitations in training samples. Even in the ResNet-time model demonstrates better performance primarily in eastern China, where the distance to monitoring sites is within 0-2 grid cells. The performance is slightly worse in the western area, where the distance to monitoring sites exceeds 4 grid cells (Figure 3d). The "-new" method, applied to the original tree-based method, also shows superior performance compared to the original, although it performs slightly worse in western China. Clearly, enhancing the training sample is crucial for further improving the model predictions, as discussed in the following.

### 3.3 Balancing site distribution is crucial to improve the prediction for the entire space of PM$_{2.5}$

Utilizing the ResNet-time model, we explore the correlation between model errors and the distance to monitor sites, as well as the concentrations at the nearest monitor. Notably, significant errors were observed in sites within a two-grid-cell distance (refer to Figure S5) and those near monitors exhibiting high concentrations (refer to Figure S6). Consequently, two criteria, namely 1) the baseline concentration in nearby monitor sites (referred to as B-conc); 2) the distance from monitor sites (referred to as D-site), are established to select the potential samples to refine predictions across the spatial domain.

Three sample groups are delineated based on the criteria:

(1) B-conc >30 µg m$^{-3}$, D-site: 1-5 grid-cell distance;

(2) B-conc within 20-30 µg m$^{-3}$, D-site: 2-5 grid-cell distance;

(3) B-conc within 10-20 µg m$^{-3}$, D-site: 3-5 grid-cell distance.

This design not only focused on the area suffering large impacts on pollution but also allowed the selection of sites in remote regions with moderate baseline concentrations, as illustrated in Figure S7. To enhance the representativeness of the chosen sites, random selections are independently conducted within each of the three groups, encompassing 10% (~300 sites, half of the existing sites), 20% (~600 sites, equal to the existing sites), 30% (~900 sites, 1.5 times the existing sites), 40% (~1200 sites), 70% (~2100 sites), and all samples (~3000 sites). The testbed developed in this study enables an efficient evaluation of the model's performance by training it with these additional sites.

The results indicate that an increase in training samples effectively enhances the model's performance in PM$_{2.5}$ estimation, with RMSE continuously decreasing as the number of samples increases (see Figure 4a). This improvement is primarily observed in the downwind area, while the performance at monitor sites deteriorates due to the original model being overfitted to these specific sites. The rate of improvement diminishes after the inclusion of 20% of samples, implying that just doubling the current ground monitors wisely can effectively balance the training samples to ensure the accuracy of PM$_{2.5}$ estimations (RMSE reduced by 20-30%). As illustrated in the example with 20% sample inclusion in Figure 4b, it is recommended that more than half of the new sites be set up in eastern China, where PM$_{2.5}$ concentration is high. Additionally, 10% of the sites are suggested to be established in remote areas that are influenced by transport from heavy pollution regions but lack nearby ground measurements. The inclusion of additional sites proves effective in significantly reducing prediction errors across the entire spatial domain, leading to a much closer agreement with the ground-truth (Figure 2a) in the PM$_{2.5}$ spatial pattern.

It is important to acknowledge that errors may be influenced by factors beyond site distribution problems, such as systematic errors arising from insufficient features. Baseline errors are referenced to those trained with all points using ResNet, amounting to within 1.7 µg m$^{-3}$ (Figure S8). Similarly, training with all points may increase errors in monitor sites, as the original model might be overfitted to these sites rather than representing the overall situation (Figure 4a).

## 3.4 Potential biases and optimized site selections under real-world conditions

While ground measurements are unavailable for the entire space, we conducted the evaluation using both the traditional RF method and the ResNet-time model developed previously with satellite data. Both models were trained using real-world satellite data and ground monitor PM$_{2.5}$ observations during 2013-2021, and their differences can be considered as part of the potential biases, associated with the influence of incorporating spatiotemporal features for enhancing the model's ability in identifying vertical structure.

The results suggest that both models effectively replicate the time series of monthly mean PM$_{2.5}$ concentrations across monitor sites from 2013 to 2021 (see Figure 5a). However, considerable disparities emerge in their predictions for other areas. The new predictions using the ResNet-time model generally exhibit lower PM$_{2.5}$ concentrations, particularly in the north and west regions (Figure 5b), with a more significant impact observed as the distance to ground monitor sites increases (Figure 5c). A notable discrepancy in population-weighted PM$_{2.5}$ concentration is observed in East China which has a large population, implying that the errors also applied in human health assessment. Since the ResNet-time model demonstrates superior performance compared to the traditional RF model in both training (reducing RMSE from 6 to 2.5 µg m$^{-3}$ and increasing R$^2$ from 0.9 to 1.0) and test data (reducing RMSE from 20 to 15 µg m$^{-3}$ and increasing R$^2$ from 0.4 to 0.6), it appears that traditional methods might significantly overestimate PM$_{2.5}$ concentrations (by 5-30 µg m$^{-3}$) and PM$_{2.5}$ exposure in suburban/rural areas by 3 million people • µg m$^{-3}$ (Figure 5c) due to the sample imbalance problem throughout 2013-2021. Similar results are also suggested in the other three benchmark models (Figure S9-S10). The actual errors might be even larger, as the inclusion of spatiotemporal-neighbourhood features in the ResNet-time model can only mitigate a portion of the errors.

Incorporating a significant number of additional sites is necessary to balance the training samples and further reduce the uncertainties. Following the two previously defined criteria (i.e., B-conc and D-site), three groups of samples are selected. Similarly, 20% of samples (631 in total, close to the number of existing sites) in each group are proposed as potential additional sites in the future, as presented in Figure 5d. Group 1 (30% of the total add-on sites) is primarily situated in polluted regions (B-conc > 60 µg m$^{-3}$, D-site: 1-5 grid-cell distance), encompassing areas such as the Beijing-Tianjin-Hebei region and the desert region in the west. Group 2 (40% of the total add-on sites) represents sites with a moderate distance to existing monitor sites with a heavier pollution level, compared to Group 1 (B-conc within 40-60 µg m$^{-3}$, D-site: 2-5 grid-cell distance). Lastly, Group 3 (30% of the total add-on sites) represents sites located far away from existing monitor sites with a low pollution level (B-conc < 40 µg m$^{-3}$, D-site: 4-5 grid-cell distance), situated in remote areas with limited influence from transport originating in pollution regions. As indicated by the previous testbed analysis, including these additional sites has

the potential to reduce errors by at least 20%, leading to a more accurate machine-learning estimation of $PM_{2.5}$ concentrations with a more balanced training sample set.

## 4 Data availability

The Numerical Model-Informed Testbed and corresponding estimated $PM_{2.5}$ concentrations spanning the past nine years (2013-2021) can be found at https://doi.org/10.5281/zenodo.11122294; and updated version with files in NetCDF format at https://zenodo.org/records/12636976. In addition to the long-term $PM_{2.5}$ dataset created from our new method, which can be used for health assessments and studying air pollution influences, we also provide testbed data crucial for evaluating ML-based retrieval methods, especially in scenarios where no ground-truth data is available.

The testbed dataset includes all inputs and outputs following the physical model simulation, which naturally correlates with physical laws such as emissions, diffusion, advection, and deposition, representing typical conditions that any prediction method should meet. This data can be used to evaluate and compare methods using the same dataset, allowing for continuous improvement. Besides traditional cross-validation, our proposed testbed validation is highly recommended to examine a method's predictive ability. We will continue updating the testbed data for other pollutants and with different resolutions and regions in future studies.

## 5 Code availability

The ResNet-time model developed in this study can be downloaded at https://doi.org/10.5281/zenodo.11122294 (Li et al., 2024).

## 6 Discussion and Conclusions

Amidst the advancements in satellite products and machine learning techniques, ground-level $PM_{2.5}$ data has found extensive applications in health assessments and related fields. However, its uncertainties have remained unexplored due to the lack of ground-truth data covering the whole space. This study designed a physically-informed testbed by leveraging the CTM simulations to evaluate $PM_{2.5}$ estimation across the entire spatial domain, quantified the associated uncertainties in the $PM_{2.5}$ mapping across the whole space. Traditionally, it was believed that errors would be significant in remote areas with few or no ground-based measurements, while observation-dense regions, following the spatial interpolation principle, were expected to exhibit better accuracy. Contrary to expectations, our findings reveal that the largest absolute biases occur differently. One reason is the heavier baseline $PM_{2.5}$ concentration, and another significant factor is the sample training imbalance problem. Ground-based measurements, designed for monitoring heavy pollution, are predominantly located in urban or industrial areas. Using these measurements as training samples misleads the machine learning model into assuming

uniform similarity to urban sites, especially in vertical structures. In reality, the vertical profile varies significantly with the flow after the pollutant is emitted from the source. This sample imbalance issue causes the machine-learning model to fail in providing accurate predictions for $PM_{2.5}$ across the entire spatial domain.

The newly developed testbed also enables us to seek best solution, such as optimizing model structure or enhancing training samples, to improve satellite-retrieval model performance. Our results underscore the importance of incorporating spatiotemporal features to enhance the machine learning model's ability to identify differences among urban and downwind conditions that are not explored in the recent literature. However, fully addressing the sample imbalance problem necessitates the addition of more ground-measurement sites to achieve a more balanced distribution of training samples for machine learning in China. In recent years, the Chinese government has expanded monitor sites towards suburban areas, increasing the total monitor sites by about 400 (from about 1600 in 2017 to about 2020 in 2021). While these additional samples have effectively improved $PM_{2.5}$ predictions (as presented in Figure S11), they account for only about 100 grid cells in the 27km by 27km domain. According to estimations in this study, approximately 600 grid cells are needed to locate monitor sites in the future. Some studies incorporate CTM simulation data as an additional feature for predicting $PM_{2.5}$, while the uncertainties of CTM hinder performance enhancement. To demonstrate that, we conducted RF predictions with CMAQ data as an additional feature, but the improvement compared to the original RF model was minimal, especially when compared to using additional neighborhood information proposed in this study (Figure S12). Besides, compared to CTM simulations, $NO_2$ column density better represents emission information and can significantly enhance model performance. As illustrated in Figure S13, excluding $NO_2$ column data from the features used in the machine learning model reduces its performance in predicting surface $PM_{2.5}$, leading to even more errors due to the sample imbalance problem.

Although this study is conducted at a relatively coarse resolution of 27 km over China due to the computational burden of running a CTM model at fine resolution on a large-scale domain, the testbed method proposed here can also be applied with higher resolution retrievals when the simulation data is available. A similar testbed study conducted in the CONUS domain at a 12-km resolution revealed the same imbalance problem (Zhang et al., in preparation), indicating that this issue persists at finer resolution scales, especially in urban and industrial areas, due to spatial heterogeneity in emissions (Li et al., 2024) and the complexity of spatial gradients of PM pollution observed at high resolution through AOD (Lin et al., 2021). At fine resolution (e.g.,1km), while the number of observation sites may increase slightly (eliminating the need for grouping to one 27km grid cell like this study), the number of grid cells to predict increases significantly. Therefore, it is valuable to conduct similar testbed studies using 1km CMAQ results (Tao et al., 2020) to evaluate the performance of ML methods. This might be more feasible with a more comprehensive CMAQ dataset using nesting over specific subdomains.

This study also successfully demonstrates leveraging the CTM model to generate abundant data for testing machine-learning methods, overcoming limitations associated with data availability. While derived from a numerical model-based testbed, it is important to acknowledge that the numerical model itself may encounter uncertainties related to emissions and chemical mechanisms, potentially leading to discrepancies with real observations. Nevertheless, the testbed serves as a specific scenario for evaluating satellite-retrieval methods, with the expectation that these methods should perform effectively in

various scenarios, including those generated from CTM simulations. Therefore, the errors observed in the CTM-based testbed also imply their existence when applied to real data. Additionally, although this study primarily focuses on the analysis of $PM_{2.5}$ in China, the identified errors may extend to other pollutants and countries as a whole, particularly when facing similar sample-imbalanced problems (i.e., lacking suburban or rural representative sites). Leveraging the testbed developed in this study can be immensely helpful in examining uncertainties in other pollutants and countries, or other geoscience applications facing similar sample imbalance challenges.

**Author contributions**

SL and JX designed the experiments and carried them out. YD helped with the data processing. JF helped with the manuscript review. SL prepared the manuscript with contributions from all co-authors.

**Competing interests**

The authors declare that they have no conflict of interest.

**Acknowledgements**

This work was supported by the Open Research Program of the International Research Center of Big Data for Sustainable Development Goals (CBAS2022ORP01), and Microsoft Climate Research Initiative program.

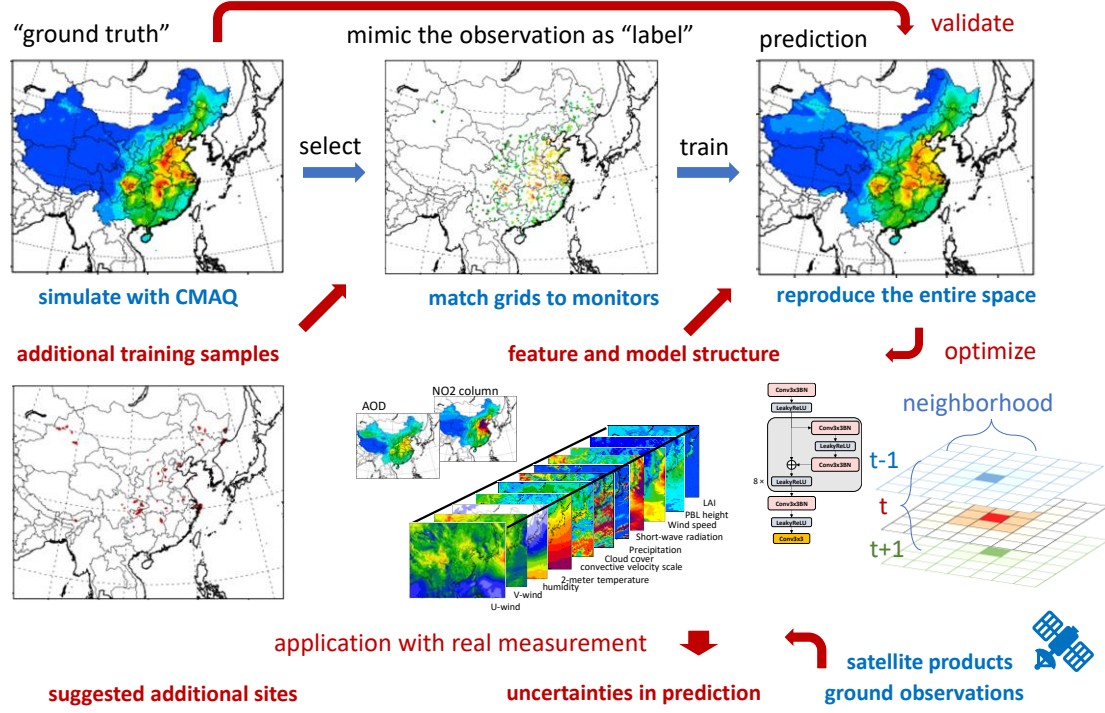

**Figure 1: The proposed testbed for evaluation of satellite-retrieving surface PM$_{2.5}$ concentration**

375

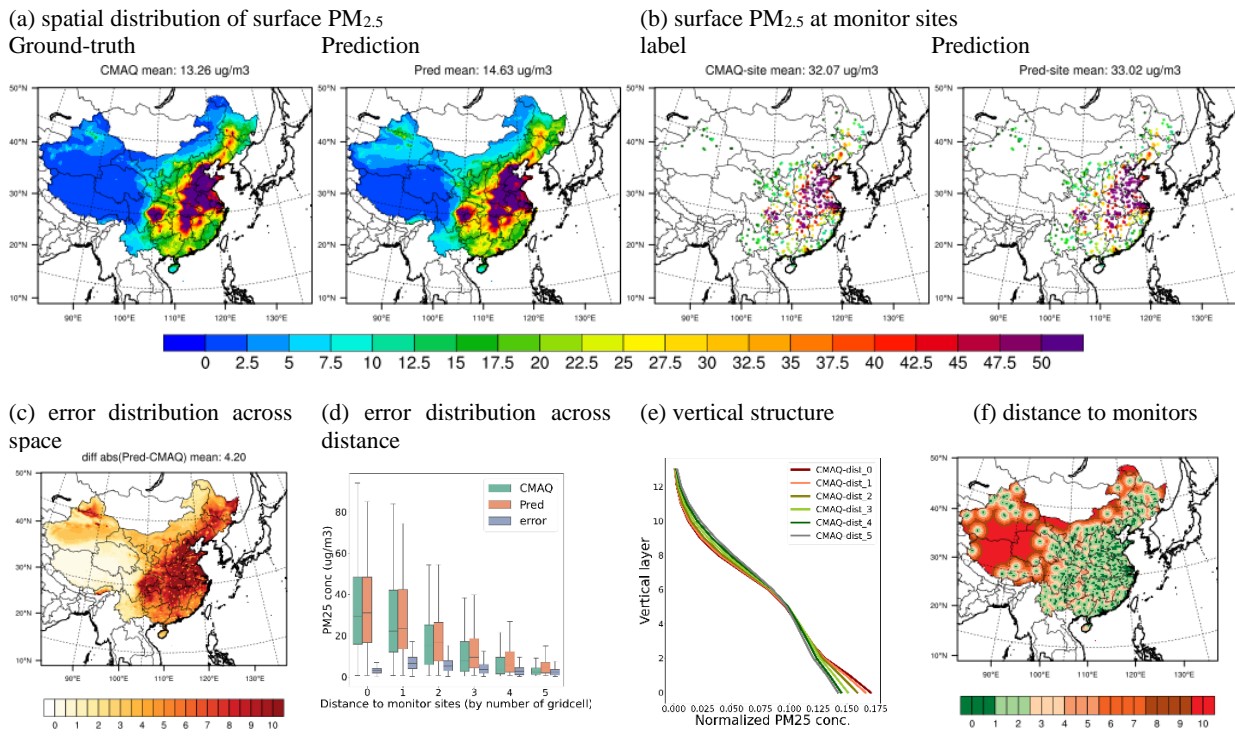

(a) spatial distribution of surface PM2.5
Ground-truth          Prediction

(b) surface PM2.5 at monitor sites
label                 Prediction

(c) error distribution across space

(d) error distribution across distance

(e) vertical structure

(f) distance to monitors

380

**Figure 2: Performance in predicting surface PM2.5 with monitor-located Random Forecast model**

(a) Prediction       (b) Performance on the sites

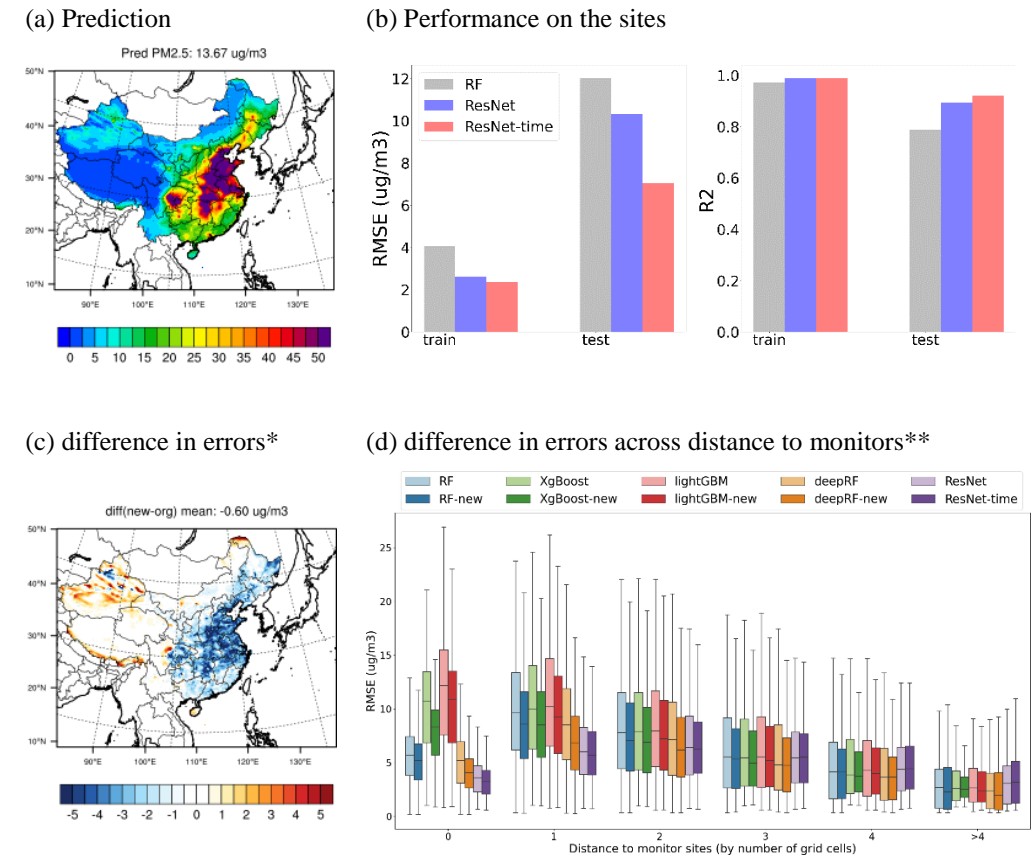

(c) difference in errors*  (d) difference in errors across distance to monitors**

Note: * blue: better; red: worse; **-new: add surrounding features into the prediction in each model

**Figure 3: Improvement after implementing the features in surrounding grid cells with ResNet (compared to RF in b and c, compared to all models in d)**

(a) performance in scenarios with adding points

overall domain

across distance to monitor sites

(b) +sample 20%

Location of sites

Prediction

difference in errors

*black dot: original monitor sites (619)

**Figure 4: Improved Performance through Integration of Additional Sites with ResNet Model**

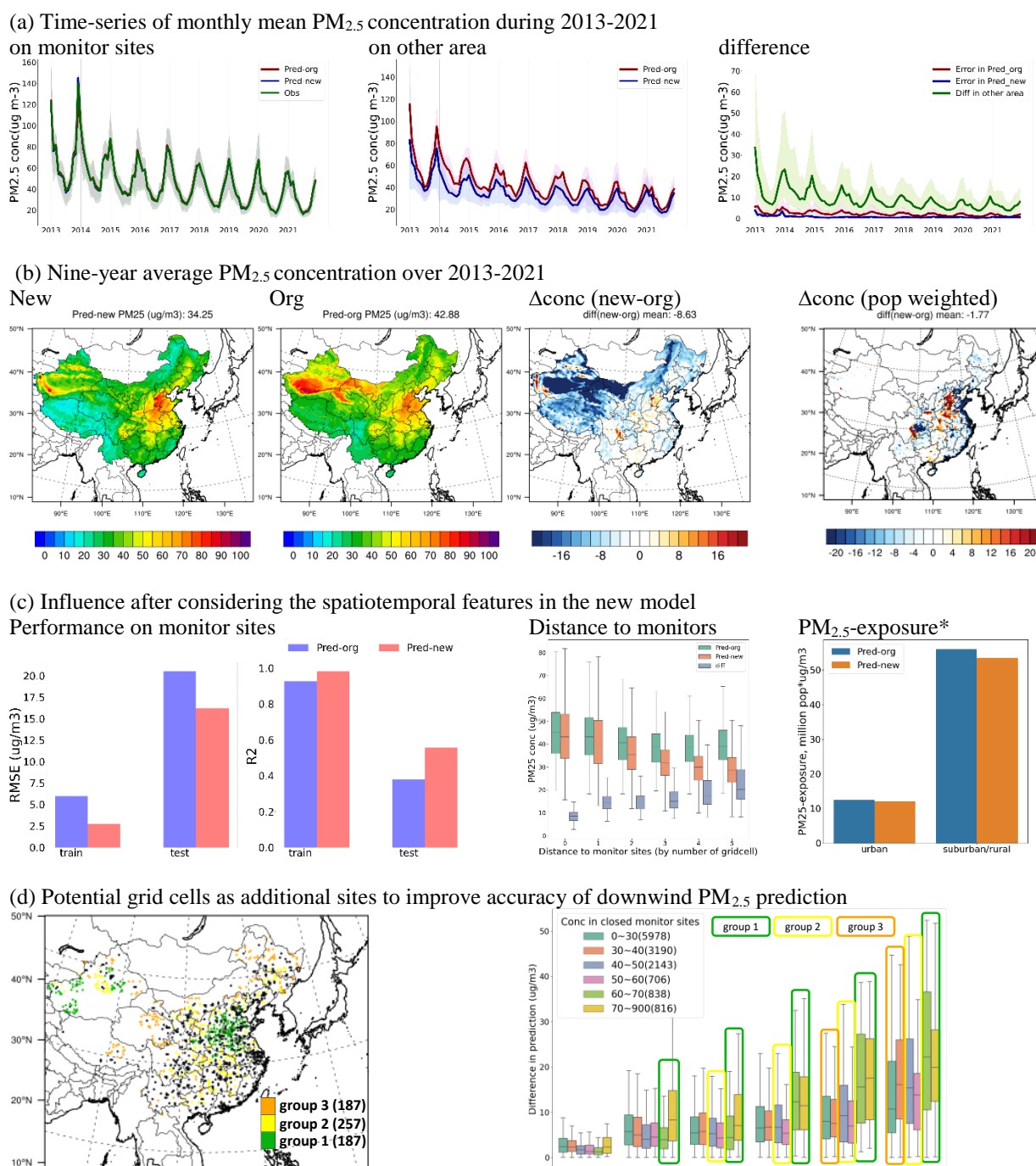

(a) Time-series of monthly mean PM$_{2.5}$ concentration during 2013-2021
on monitor sites      on other area      difference

(b) Nine-year average PM$_{2.5}$ concentration over 2013-2021
New      Org      Δconc (new-org)      Δconc (pop weighted)

(c) Influence after considering the spatiotemporal features in the new model
Performance on monitor sites      Distance to monitors      PM$_{2.5}$-exposure*

(d) Potential grid cells as additional sites to improve accuracy of downwind PM$_{2.5}$ prediction

**Figure 5: Improve the estimation of PM$_{2.5}$ and related exposure across China with satellite product and ground observations during 2013-2021 (*suburban/rural represent the area within 5 grid cells distance to the monitor sites; results are comparing the ResNet-time model with the baseline RF)**

395

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
