# Peer review of "Retrieving Ground-Level PM2.5 Concentrations in China (2013-2021) with a Numerical Model-Informed Testbed to Mitigate Sample Imbalance-Induced Biases"

_Earth System Science Data, 2024_

## Author Response (AR1)

We appreciate the reviewers' valuable comments and constructive suggestions which help us improve the quality of the manuscript. We have carefully revised the manuscript according to these comments. Point-to-point responses are provided below. The reviewers' comments are in black, our responses are in blue with the detailed changes in the revised manuscript and labeled with the Page and Line number of the revised manuscript highlighted the corresponding changes at the end in red.

**Reviewer: 1**

Comments:

This is an important paper for the community of air quality estimates from satellite remote sensing and utilizing machine learning, by assessing uncertainties due to the placement of ground-based observations in the gapless PM2.5 estimates based on such method. In doing so, the authors use synthetically modeled PM2.5 from a state-of-art high-resolution air quality model in China, and test the biases of PM2.5 estimates using the current site placement. The data processing, assumptions, and evaluation logics are clearly described. The analysis is overall sound and described in details. The findings are important and implicative for the development of PM2.5 estimates, and for future placement of PM2.5 monitoring sites in China and other polluted regions of the world. More work is needed to improve the applicability of the derived data, especially regarding the journal ESSD. I support the publication of this manuscript provided that the following comments can be addressed.

[Response]

We thank the reviewer for assessing the manuscript and recognition of the implications of the results of the analysis presented, and overall positive comments. We have followed all the comments and revised manuscript accordingly.

Major comments:

1) The main conclusion "Larger PM2.5 biases found in densely populated regions" needs more consideration. This statement is based on ABSOLUTE biases. Meanwhile, these regions associate with high PM2.5, so larger absolute PM2.5 biases are expected. Normalized (concentration-independent) biases statistics should also be provided to better discuss the results and guide the analysis.

[Response]

We agree with the reviewer that our original statement was somewhat misleading. Our intention was to address that the problem can still be very significant in polluted regions due to the high baseline concentration. As the reviewer suggested, we estimated the relative error (normalized to the baseline concentration), as shown in the figure below. Clearly, the relative error is significantly larger in western China, where the baseline concentration (denominator) is lower. Conversely, in the polluted regions of

eastern China, the original RF model shows errors of around 20%. However, these errors are significantly reduced with the implementation of neighborhood information in the RF-new and ResNet-time models.

We have revised the statement and made it clearer in the revised manuscript following the reviewer's suggestion as follows.

(Page 1 Line 19) "Unexpected results are shown in the application in China, with larger absolute PM$_{2.5}$ biases found in densely populated regions with abundant ground observations across all benchmark models due to the higher baseline concentration, though relative errors (approximately 20%) is smaller compared to rural areas (over 50%)"

(Page 6 Line 186) "Contrary to traditional expectations, significant absolute errors occur mostly in East China rather than in the West due to the large baseline concentration, even though the relative error is smaller (about 20%) (Figure S2)."

(Page 7 Line 219) " The incorporation of surrounding features in the ResNet-time model significantly mitigates both absolute and relative errors in East China across the spatial domain (see Figure 3c; Figure S2)."

(Page 10 Line 320) " Contrary to expectations, our findings reveal that the largest absolute biases occur differently."

[Figure]

**Figure S2.** Comparison of relative errors in predicting ground-level PM$_{2.5}$ using RF, RF-new, and ResNet-time models with the testbed

2) It is unclear in the current manuscript, if the input data (predictors of PM2.5) are identical between the tree-based methods and the RedNet in Section 2. To make a fair comparison, please clarity this issue. If they are not consistent, please explain why.

[Response]

The inputs for all models are exactly the same to ensure a fair comparison. The only difference is that the features for the new proposed methods include data from the neighborhood (nearby grid cells and

previous/next time-steps) in addition to the local grid and time data. We have clarified this point in the revised manuscript as follows:

(Page 5 Line 162) "One thing should be noted that all machine learning methods use the same input features to ensure a fair comparison. Only difference is that the features for the new proposed methods (e.g., ResNet) include data from the neighborhood (nearby grid cells and previous/next time-steps) in addition to the local grid and time data."

3) The machine-learning estimation performed in this paper are annual averages at 27 km resolution. PM2.5 air quality applications usually require higher spatial resolution due to its strong heterogeneity and co-variability with population. For an ESSD paper, please add a paragraph to explain and discuss the potential applications of such data, since such applications are unclear to me. Also, the current form of data (in python .npy format) is very hard to use, and the data description in the README cannot support the users to use the data. Please consider changing them to more widely used format (e.g., netcdf) with geo-references, and providing more detailed descriptions. If the applicability of the generated data is not well justified and improved, I personally think this paper suits better for other journals like ACP, but I am open to leave the judgement of suitability to the editor.

[Response]

We understand the reviewer's concern about the usage of this data. One purpose of this paper is to provide a long-term PM2.5 dataset for China, created using our new method. Another important contribution is the testbed dataset and code, which are crucial for evaluating ML-based retrieval methods, particularly in scenarios where there is no ground-truth data available for validation.

The testbed dataset includes all inputs and outputs from physical model simulations, which adhere to physical laws such as emissions, diffusion, advection, and deposition. This dataset represents typical conditions that any prediction method should meet. We submitted our work to ESSD to share this unique testbed data with the community, enabling the evaluation and comparison of methods using the same dataset and fostering continuous improvement.

In addition to traditional cross-validation, we highly recommend testbed validation to examine a method's predictive ability. We also plan to continue updating the testbed data for other pollutants and with different resolutions and regions in future studies.

Following the reviewer's suggestion, we also re-created the data to netcdf format with geo-references and upload online to share with the community.

We clarify this point in the revised manuscript as follows.

(Page 10 Line 299) "The Numerical Model-Informed Testbed and corresponding estimated PM2.5 concentrations spanning the past nine years (2013-2021) can be found at https://doi.org/10.5281/zenodo.11122294; and updated version with files in NetCDF format at https://zenodo.org/records/12636976. In addition to the long-term PM$_{2.5}$ dataset created from our new method, which can be used for health assessments and studying air pollution influences, we also provide testbed data crucial for evaluating ML-based retrieval methods, especially in scenarios where no ground-truth data is available.

The testbed dataset includes all inputs and outputs following the physical model simulation, which naturally correlates with physical laws such as emissions, diffusion, advection, and deposition, representing typical conditions that any prediction method should meet. This data can be used to evaluate and compare methods using the same dataset, allowing for continuous improvement. Besides traditional cross-validation, our proposed testbed validation is highly recommended to examine a method's predictive ability. We will continue updating the testbed data for other pollutants and with different resolutions and regions in future studies."

Other comments:

1) Line 19-21: As outlined before, I believe relative errors should be discussed apart from absolute errors. The current statement might be misleading.

[Response]

As the reviewer suggested, we have clarified this point in the revised manuscript as follows.

(Page 1 Line 19) "Unexpected results are shown in the application in China, with larger absolute $PM_{2.5}$ biases found in densely populated regions with abundant ground observations across all benchmark models due to the higher baseline concentration, though relative errors (approximately 20%) is smaller compared to rural areas (over 50%)."

2) Line 64: Based on my experiences with the PM2.5 data in China, many sites can be apart from each other within <20 km. How did you deal with a 27-km grid cell containing >1 site? Furthermore, how would the 27 km model resolution affect the evaluation of conventional PM2.5 estimation approach, considering that these existing data in literature are estimated at finer (e.g., 1km) resolution? Overall, more discussion about model vs. desirable resolution, and how the insights from this paper could guide evaluation of PM2.5 estimates at finer resolution should be provided.

[Response]

We average the concentrations for sites located within the same 27-km grid cells to better represent the overall concentration at this resolution. The testbed can also be conducted with higher resolution retrievals when the simulation data is available. A similar testbed study conducted in the US at a 12-km resolution revealed a similar imbalance problem (Zhang et al., in preparation), indicating that this issue persists from 27-km to 12-km resolutions.

Due to limitations of the CMAQ simulations, high resolutions such as 1-km can only be applied to smaller domains, as previously done for the Beijing area (Tao et al., 2021). It is expected that similar sample-imbalance problems will occur, as the number of observation sites increases slightly (eliminating the need for grouping), while the number of grid cells increases significantly with higher resolution.

Moreover, the sample imbalance problem persists, especially in urban and industrial areas due to spatial heterogeneity in emissions (Li et al., 2024) and the complexity of spatial gradients of PM pollution observed at high resolution through AOD (Lin et al., 2022). Therefore, it is still valuable to conduct

similar testbed studies using 1-km CMAQ to evaluate the performance of ML methods. This might be more feasible with a more comprehensive CMAQ dataset using nesting over specific subdomains.

We clarified this point in the revised manuscript as follows.

(Page 11 Line 343) "Although this study is conducted at a relatively coarse resolution of 27 km over China due to the computational burden of running a CTM model at fine resolution on a large-scale domain, the testbed method proposed here can also be applied with higher resolution retrievals when the simulation data is available. A similar testbed study conducted in the CONUS domain at a 12-km resolution revealed the same imbalance problem (Zhang et al., in preparation), indicating that this issue persists at finer resolution scales, especially in urban and industrial areas, due to spatial heterogeneity in emissions (Li et al., 2024) and the complexity of spatial gradients of PM pollution observed at high resolution through AOD (Lin et al., 2021). At fine resolution (e.g.,1km), while the number of observation sites may increase slightly (eliminating the need for grouping to one 27km grid cell like this study), the number of grid cells to predict increases significantly. Therefore, it is valuable to conduct similar testbed studies using 1km CMAQ results (Tao et al., 2020) to evaluate the performance of ML methods. This might be more feasible with a more comprehensive CMAQ dataset using nesting over specific subdomains."

Reference:

Li, S., & Xing, J. (2024). DeepSAT4D: Deep learning empowers four-dimensional atmospheric chemical concentration and emission retrieval from satellite. *The Innovation Geoscience*, *2*(1), 100061-1.

Lin, H., Li, S., Xing, J., He, T., Yang, J., & Wang, Q. (2021). High resolution aerosol optical depth retrieval over urban areas from Landsat-8 OLI images. *Atmospheric Environment*, *261*, 118591.

Tao, H., Xing, J., Zhou, H., Pleim, J., Ran, L., Chang, X., Wang, S., Chen, F., Zheng, H. and Li, J., 2020. Impacts of improved modeling resolution on the simulation of meteorology, air quality, and human exposure to PM2. 5, O3 in Beijing, China. *Journal of Cleaner Production*, *243*, p.118574.

3) Line 125: Should firstly discuss the importance of representing varying aerosol vertical profiles in source and downwind areas in the Introduction Section.

[Response]

As the reviewer suggested, we have added this description in the introduction section, as follows.

(Page 2 Line 50) "The discrepancy between urban and downwind sites largely lies in their vertical profiles of aerosol across the entire vertical layers. Urban sites, which have abundant emission sources such as residential areas, transportation, construction, and industries, exhibit a higher share of ground-level aerosol relative to the total AOD compared to downwind and rural areas, where pollution tends to be lifted to upper layers through atmospheric dynamics. Accurately representing the varying aerosol vertical profiles in source/urban and downwind/rural areas is crucial for retrieving ground-level $PM_{2.5}$

from the AOD. However, imbalanced training samples make the machine-learning model unable to adequately capture such variations. The traditional cross-validation methods either based on samples or sites (Dong et al., 2020), which still rely mostly on samples available in urban sites, fails to provide a comprehensive assessment of model performance across the entire prediction space."

Reference:

Dong, L., Li, S., Yang, J., Shi, W., & Zhang, L. (2020). Investigating the performance of satellite-based models in estimating the surface PM2. 5 over China. *Chemosphere*, *256*, 127051.

4) Section 3.2 and Figure 3: The tree-based methods seem showing improved performance after adding surrounding features, and the "-new" RMSE values are comparable to ResNet-time. For example, the xgboost-new RMSE looks better than ResNet-time at distance >2 grids. Overall, I do not agree that the ResNet and ResNet-time approaches are that overwhelmingly superior. Please more accurately discuss these features and revise the evaluation of these approaches accordingly. Also, why are the RF-new or XgBoost-new results not shown on Figure 3b and 3c?

[Response]

We agree with the reviewer that the original statement was somewhat arbitrary. The ResNet-time model demonstrates better performance primarily in eastern China, where the distance to monitoring sites is within 0-2 grid cells. The performance is slightly worse in the western area, where the distance to monitoring sites exceeds 4 grid cells. The "-new" method, applied to the original tree-based method, also shows superior performance compared to the original, although it performs slightly worse in western China (as shown in Figure S3). The results for other models with the "-new" method are compared in Figure S3.

Following the reviewer's suggestion, we have updated the discussion about performance in the revised manuscript as follows:

(Page 7 Line 229) "However, performance is barely improved or even worsens in faraway sites (distance >4 grid cells) due to the limitations in training samples. Even in the ResNet-time model demonstrates better performance primarily in eastern China, where the distance to monitoring sites is within 0-2 grid cells. The performance is slightly worse in the western area, where the distance to monitoring sites exceeds 4 grid cells (Figure 3d). The "-new" method, applied to the original tree-based method, also shows superior performance compared to the original, although it performs slightly worse in western China."

**Reviewer: 2**

Comments:

The authors generated a new PM2.5 concentration dataset in China with a new PM2.5 modeling framework, aiming at mitigating sample imbalance-induced biases. The topic is of interest and is important for improving satellite-based PM2.5 modeling. The following flaws should be addressed to improve the quality of this manuscript.

[Response]

We thank the reviewer for assessing the manuscript and recognition of the implications of the results of the analysis presented, and overall positive comments. We have followed all the comments and revised manuscript accordingly.

1. The writing style of the introduction section is more likely a technical report since the authors used three paragraphs to describe the modeling method.
[Response]

As the reviewer suggested, we reorganized the introduction section of the manuscript, extending it slightly, and moved some discussion about the modeling method into the Method sections.

(Page 2 Line 50) "The discrepancy between urban and downwind sites largely lies in their vertical profiles of aerosol across the entire vertical layers. Urban sites, which have abundant emission sources such as residential areas, transportation, construction, and industries, exhibit a higher share of ground-level aerosol relative to the total AOD compared to downwind and rural areas, where pollution tends to be lifted to upper layers through atmospheric dynamics. Accurately representing the varying aerosol vertical profiles in source/urban and downwind/rural areas is crucial for retrieving ground-level PM$_{2.5}$ from the AOD. However, imbalanced training samples make the machine-learning model unable to adequately capture such variations. The traditional cross-validation methods either based on samples or sites (Dong et al., 2020), which still rely mostly on samples available in urban sites, fails to provide a comprehensive assessment of model performance across the entire prediction space."

2. Section 2.1: the authors stated that the emission data of ABaCAS-EI has a spatial resolution of 1km-by-1km and a temporal resolution of 1 hour. Please double check the resolution. As reported by the data producer, the temporal resolution of this dataset is annual.
[Response]

We thank the reviewer for pointing out this issue. ABaCAS-EI was processed using fine-scale surrogate data such as population and road maps, which can be mapped to 1 km by 1 km grid cells. In our previous

paper (Tao et al.,2020), we refined the emissions to this resolution by processing the spatial allocation of emissions with additional proxy data to match the 1 km resolution in air quality modeling. Sector emissions with precise spatial locations, such as those from power generation, iron, steel, cement, non-ferrous metal production, and coking, are treated as point sources. Emissions from transportation, other industries, and domestic sources are allocated based on economic and population data, as well as high-resolution land-use data, to avoid arbitrarily allocating emissions solely to urban areas with high population density. The dataset may not be publicly available yet.

However, in this study, since our CMAQ model was conducted at 27km resolution, to avoid confusing, we rewrote the statement in the revised manuscript as follows:

(Page 3 Line 96) "Emission data were obtained from the high-resolution emission inventory developed by Tsinghua University (ABaCAS-EI) (Zheng et al., 2019), characterized by a spatial resolution as 27km-by-27km and a temporal resolution of 1 hour to match with the CMAQ model"

Reference

Tao, H., Xing, J., Zhou, H., Pleim, J., Ran, L., Chang, X., Wang, S., Chen, F., Zheng, H. and Li, J., 2020. Impacts of improved modeling resolution on the simulation of meteorology, air quality, and human exposure to PM2. 5, O3 in Beijing, China. Journal of Cleaner Production, 243, p.118574.

3. Line 99-100: "The remaining grid cells encompass the surrounding PM2.5, which has not been previously evaluated in other studies", the validation schemes for PM2.5 had been intercompared in previous studies, also including how to evaluate the model's extrapolation capacity.

[Response]

We appreciate the reviewer pointing out this statement, which was somewhat confusing. Here, we intend to convey that previous studies have explored validation methods such as cross-validation by sites, samples, or time for evaluating the model's extrapolation capacity. However, due to the lack of ground truth data for regions outside monitoring sites, previous studies were unable to directly assess prediction errors against observations (due to their unavailability) for those grid cells. We have clarified this point in the revised manuscript as follows:

(Page 4 Line 109) "Though previous studies provide some validation schemes to evaluate the model's extrapolation capacity (Dong et al., 2020), for the remaining grid cells that encompass surrounding $PM_{2.5}$ areas where observations are not available, the predicted concentrations with machine learning method cannot be directly compared and examined."

4. How was the accuracy of the CTM results in rural areas (without CNEMC monitors) evaluated?

[Response]

Since there are no observations available in rural areas, we cannot directly compare the CTM predictions with actual observations. However, the CTM data used in this study serves primarily to establish a testbed representing a scenario based on physical laws such as emission, diffusion, advection, and deposition. This approach differs significantly from reanalysis or data fusion methods, which may not strictly adhere to these physical functions, even though they might show better agreement with observations when available.

Due to these issues, the uncertainties of CTM predictions also limit its utility as a feature for machine learning methods. While some studies incorporate CTM simulation data as an additional feature for predicting PM2.5, its uncertainties hinder performance enhancement, as demonstrated in this study. We conducted RF predictions with CTM data as an additional feature, but the improvement compared to the original RF model was minimal, especially when compared to using additional neighborhood information proposed in this study (i.e., RF-new). This suggests that CTM's limited utility as an additional feature for addressing the sample imbalance problem is compounded by its inherent uncertainties.

We have clarified this point in the revised manuscript as follows.

(Page 4 Line 115) "The WRF/CMAQ simulations were evaluated in our previous studies (Ding et al., 2019ab), demonstrating acceptable agreement with CNEMC observations, albeit with limitations in areas where observations are available. In rural areas where no observations are available, direct comparison of CMAQ predictions with actual observations is not possible. However, the CMAQ data used in this study primarily serves to establish a testbed representing scenarios based on physical laws such as emission, diffusion, advection, and deposition. This approach contrasts with reanalysis or data fusion methods, which may deviate from these physical functions, even though they might exhibit better agreement with observations when available."

(Page 11 Line 336) "Some studies incorporate CTM simulation data as an additional feature for predicting PM2.5, while the uncertainties of CTM hinder performance enhancement. To demonstrate that, we conducted RF predictions with CMAQ data as an additional feature, but the improvement compared to the original RF model was minimal, especially when compared to using additional neighborhood information proposed in this study (Figure S12)."

[Figure]

Figure S12 Comparison of predicted PM2.5 by adding simulation data and the proposed method in this study (2017 for example)

Reference:

Ding, D., Xing, J., Wang, S., Chang, X., Hao, J., 2019a. Impacts of emissions and meteorological changes on China's ozone pollution in the warm seasons of 2013 and 2017. Frontiers of Environmental Science & Engineering, 13(5), 1-9.

Ding, D.; Xing, J.; Wang, S.; Liu, K.; Hao, J., 2019b. Estimated Contributions of Emissions Controls, Meteorological Factors, Population Growth, and Changes in Baseline Mortality to Reductions in Ambient PM2.5 and PM2.5 -Related Mortality in China, 2013-2017., Environ Health Perspect., 127(6):67009.

5. The NO2 column density from satellite suffers from significant data gaps, how did the authors account for this issue in their study?

[Response]

We appreciate the reviewer for highlighting this issue. In our testbed case, we do not encounter data gaps since all data are derived from continuous simulation data. However, when applying our approach to real data, we first perform data filling for satellite data using traditional methods before proceeding with further analysis. We acknowledge the importance of data filling for satellite retrievals as emphasized by the reviewer. In this study, our primary focus is on addressing challenges related to machine learning methods rather than addressing missing values in satellite data. This distinction is important because even with continuous satellite data, we may still encounter the same sample-imbalance problem.

We have clarified this point in the revised manuscript as follows.

(Page 3 Line 87) "Following the same data filling method (He et al., 2020), we conduct data filling for the satellite measurement of $NO_2$ column density and AOD when applying our approach to real data."

Reference:

He, Q.; Qin, K.; Cohen, J. B.; Loyola, D.; Li, D.; Shi, J.; Xue, Y. Spatially and temporally coherent reconstruction of tropospheric NO2 over China combining OMI and GOME-2B measurements. Environ. Res. Lett. 2020, 15, 125011.

6. Line 120-121: "the model is trained using data from only the first 25 days of each month,", why didn't choose randomly?
[Response]

The selection of the first 25 days of each month simply because we want to make sure all the method is using exactly the same data for training and testing, to make a fair comparison. If we select randomly, we still need to fixed the randomly selected days for all methods, which is quite similar as fixed all the days at the beginning. Moreover, the purpose of this study is to investigate the error in prediction for

grid cells that not included in the training dataset, even for the first 25 days, we still see similar errors in prediction in other sites (all like out-of-sample validation) no matter which days select for training or testing. We clarified this point in the revised manuscript as follows.

(Page 5 Line 142) "We fixed the days for training rather than selecting them randomly to ensure that all methods use exactly the same data for training and testing, enabling a fair comparison. Random selection would still require fixing the randomly selected days for all methods, similar to fixing all days at the outset. Moreover, the purpose of this study is to investigate prediction errors for grid cells not included in the training dataset. Even when using the first 25 days of training dataset, we consistently observe similar prediction errors in other sites (similar to out-of-sample validation), regardless of which days are selected for training or testing."

7. What parameters were used in machine learning models? In section 2.2 the authors mentioned meteorological factors, land use, and NO2 density. However, in section 2.3, the authors mentioned "Beyond simply including corresponding features from the surrounding neighborhood grid cells as additional predictors for predicting PM2.5 concentration at the target grid cells in decision tree-based methods". The whole logic flow is a bit confusing, which needs to be improved to ease the readership.
[Response]

The inputs for all models are exactly the same to ensure a fair comparison. The only difference is that the features for the new proposed methods include data from the neighborhood (nearby grid cells and previous/next time-steps) in addition to the local grid and time data. We have clarified this point in the revised manuscript as follows:

(Page 5 Line 162) "One thing should be noted that all machine learning methods use the same input features to ensure a fair comparison. Only difference is that the features for the new proposed methods (e.g., ResNet) include data from the neighborhood (nearby grid cells and previous/next time-steps) in addition to the local grid and time data."

8. Line 156: "Therefore, the model, trained with urban sites, attributes more pollution to the ground level from the AOD", was AOD used as an explanatory variable?
[Response]

The AOD is one of the feature variables used for prediction, contributed by each vertically layers, in which the ground level is the one for the target. Ideally, the ML-model aims to estimate the ground level PM2.5, which can be regarded as a ratio from the total AOD for the ground-level, by using all the features fed in the ML model. In this sentence, we'd like to explain, if we trained with urban sites where the ratio is mostly higher than rural area, the linkage between it with other features might be not suitable for rural/downwind area.

To clarify this point also to emphasis the importance of the varying ratio of ground-level to total AOD, we revised this sentence and added some discussion in the introduction section in the revised manuscript as follows.

(Page 2 Line 50) "The discrepancy between urban and downwind sites largely lies in their vertical profiles of aerosol across the entire vertical layers. Urban sites, which have abundant emission sources such as residential areas, transportation, construction, and industries, exhibit a higher share of ground-level aerosol relative to the total AOD compared to downwind and rural areas, where pollution tends to be lifted to upper layers through atmospheric dynamics. Accurately representing the varying aerosol vertical profiles in source/urban and downwind/rural areas is crucial for retrieving ground-level PM$_{2.5}$ from the AOD. However, imbalanced training samples make the machine-learning model unable to adequately capture such variations. The traditional cross-validation methods either based on samples or sites (Dong et al., 2020), which still rely mostly on samples available in urban sites, fails to provide a comprehensive assessment of model performance across the entire prediction space."

(Page 6 Line 183) "Therefore, the model, trained on urban sites where ground-level pollution from AOD is more prominent, failed to accurately capture the diverse aerosol vertical profiles in source/urban and downwind/rural areas. This discrepancy resulted in overestimations in downwind areas."

9. Line 165-166: "we can conclude that the uneven distribution of sites introduces considerable biases in PM2.5 estimation within traditional methods that rely on local features.", previous machine-learned PM2.5 modeling relies largely on satellite AOD. The method used in this study relies largely on NO2 density, which is pretty high in urban and low in rural. Is this a possible reason for causing such modeling difference?

[Response]

The ML method will leverage the feature information based on their similarity. If the feature has large contribution to the performance, it will be used heavily. The model we present in this study relies largely on NO2 density simply because it has usefully indicator for the emission sources, also an important precursor for PM2.5 secondary formation. To address the reviewer's concern, we also conduct additional experiment that without using NO2, and the performance getting worse, also suffering more significant imbalance problem. We describe it in the revised manuscript as follows.

(Page 11 Line 339) "Besides, compared to CTM simulations, NO$_2$ column density better represents emission information and can significantly enhance model performance. As illustrated in Figure S13, excluding NO$_2$ column data from the features used in the machine learning model reduces its performance in predicting surface PM$_{2.5}$, leading to even more errors due to the sample imbalance problem."

[Figure]

Figure 13. Comparison of model performance in predicting surface PM$_2.5$ without NO$_2$ column feature

10. The dataset presented in the study is good for evaluating existing or any new method for PM2.5 retrievals from satellites, while its spatial resolution is only 27km by 27km. Can it be also applied for studies using high resolution dataset?

[Response]

We average the concentrations for sites located within the same 27-km grid cells to better represent the overall concentration at this resolution. The testbed can also be conducted with higher resolution retrievals when the simulation data is available. A similar testbed study conducted in the US at a 12-km resolution revealed a similar imbalance problem (Zhang et al., in preparation), indicating that this issue persists from 27-km to 12-km resolutions.

Due to limitations of the CMAQ simulations, high resolutions such as 1-km can only be applied to smaller domains, as previously done for the Beijing area (Tao et al., 2021). It is expected that similar sample-imbalance problems will occur, as the number of observation sites increases slightly (eliminating the need for grouping), while the number of grid cells increases significantly with higher resolution.

Moreover, the sample imbalance problem persists, especially in urban and industrial areas due to spatial heterogeneity in emissions (Li et al., 2024) and the complexity of spatial gradients of PM pollution observed at high resolution through AOD (Lin et al., 2022). Therefore, it is still valuable to conduct similar testbed studies using 1-km CMAQ to evaluate the performance of ML methods. This might be more feasible with a more comprehensive CMAQ dataset using nesting over specific subdomains.

We clarified this point in the revised manuscript as follows.

(Page 11 Line 343) "Although this study is conducted at a relatively coarse resolution of 27 km over China due to the computational burden of running a CTM model at fine resolution on a large-scale domain, the testbed method proposed here can also be applied with higher resolution retrievals when the simulation data is available. A similar testbed study conducted in the CONUS domain at a 12-km resolution revealed the same imbalance problem (Zhang et al., in preparation), indicating that this issue persists at finer resolution scales, especially in urban and industrial areas, due to spatial heterogeneity in emissions (Li et al., 2024) and the complexity of spatial gradients of PM pollution observed at high resolution through AOD (Lin et al., 2021). At fine resolution (e.g.,1km), while the number of observation sites may increase slightly (eliminating the need for grouping to one 27km grid cell like this study), the number of grid cells to predict increases significantly. Therefore, it is valuable to conduct similar testbed studies using 1km CMAQ results (Tao et al., 2020) to evaluate the performance of ML methods. This might be more feasible with a more comprehensive CMAQ dataset using nesting over specific subdomains."

Reference:

Li, S., & Xing, J. (2024). DeepSAT4D: Deep learning empowers four-dimensional atmospheric chemical concentration and emission retrieval from satellite. *The Innovation Geoscience*, *2*(1), 100061-1.

Lin, H., Li, S., Xing, J., He, T., Yang, J., & Wang, Q. (2021). High resolution aerosol optical depth retrieval over urban areas from Landsat-8 OLI images. *Atmospheric Environment*, *261*, 118591.

Tao, H., Xing, J., Zhou, H., Pleim, J., Ran, L., Chang, X., Wang, S., Chen, F., Zheng, H. and Li, J., 2020. Impacts of improved modeling resolution on the simulation of meteorology, air quality, and human exposure to PM2. 5, O3 in Beijing, China. *Journal of Cleaner Production*, *243*, p.118574.